# Host nutritional status affects alphavirus virulence, transmission, and evolution

**James Weger-Lucarelli**[1,2¤]*, **Lucia Carrau**[1], **Laura I. Levi**[1,3], **Veronica Rezelj**[1], **Thomas Vallet**[1], **Hervé Blanc**[1], **Jérémy Boussier**[4], **Daniela Megrian**[5], **Sheryl Coutermarsh-Ott**[2], **Tanya LeRoith**[2], **Marco Vignuzzi**[1]

1 Institut Pasteur, Viral Populations and Pathogenesis Unit, Paris, France, 2 Department of Biomedical Sciences and Pathobiology, Virginia Tech, VA-MD Regional College of Veterinary Medicine, Blacksburg, VA, United States of America, 3 Ecole doctorale BioSPC, Université Paris Diderot, Sorbonne Paris Cité, Paris, France, 4 Institut Pasteur, Immunobiology of Dendritic Cells, Institut National de la Santé et de la Recherche Médicale, Paris, France, 5 Institut Pasteur, Evolutionary Biology of the Microbial Cell, Department of Microbiology, Paris, France

¤ Current address: Department of Biomedical Sciences and Pathobiology, Virginia Tech, VA-MD Regional College of Veterinary Medicine, Blacksburg, VA, United States of America
* weger@vt.edu

**Data Availability Statement:** All next-generation sequencing files are uploaded to the small read archive (SRA) under accession number PRJNA573904.

## Abstract

Malnourishment, specifically overweight/obesity and undernourishment, affects more than 2.5 billion people worldwide, with the number affected ever-increasing. Concurrently, emerging viral diseases, particularly those that are mosquito-borne, have spread dramatically in the past several decades, culminating in outbreaks of several viruses worldwide. Both forms of malnourishment are known to lead to an aberrant immune response, which can worsen disease outcomes and reduce vaccination efficacy for viral pathogens such as influenza and measles. Given the increasing rates of malnutrition and spread of arthropod-borne viruses (arboviruses), there is an urgent need to understand the role of host nutrition on the infection, virulence, and transmission of these viruses. To address this gap in knowledge, we infected lean, obese, and undernourished mice with arthritogenic arboviruses from the genus Alphavirus and assessed morbidity, virus replication, transmission, and evolution. Obesity and undernourishment did not consistently influence virus replication in the blood of infected animals except for reductions in virus in obese mice late in infection. However, morbidity was increased in obese mice under all conditions. Using Mayaro virus (MAYV) as a model arthritogenic alphavirus, we determined that both obese and undernourished mice transmit virus less efficiently to mosquitoes than control (lean) mice. In addition, viral genetic diversity and replicative fitness were reduced in virus isolated from obese compared to lean controls. Taken together, nutrition appears to alter the course of alphavirus infection and should be considered as a critical environmental factor during outbreaks.

## Author summary

Over- and undernutrition, collectively known as malnutrition, affect over 2.5 billion people worldwide. Associations between malnutrition and mosquito-borne virus infection

**Funding:** This work was partially funded by the DARPA program PREventing EMerging Pathogenic Threats (PREEMPT) awarded to MV and JWL. Partial funding was also provided by a faculty start-up package at Virginia Tech awarded to JWL. The funders had no role in study design, data collection and analysis, decision to publish, or preparation of the manuscript

**Competing interests:** The authors have declared that no competing interests exist.

and resulting disease have been identified in epidemiological studies but have not been explored in controlled studies. Here, we infect obese or undernourished mice with different arthritis inducing viruses in the genus *Alphavirus* and measure disease symptoms, viral replication, transmission, and evolution. We found that markers of disease, namely weight loss and footpad swelling, were increased in obese mice. We also found that replication differences between mice fed different diets were minimal except late in infection for obese mice when levels of virus dropped significantly. When mosquitoes were allowed to feed on mice fed different diets, we observed reduced infection and transmission rates, depending on the diet. Finally, we found reduced genetic diversity and replicative fitness of virus isolated from obese mice. This study provides insights into the influence of nutrition on alphavirus pathogenesis and evolution.

## Introduction

Arthritogenic alphaviruses are globally distributed and have been responsible for several recent large outbreaks. Notably, chikungunya virus (CHIKV) re-emerged in 2004 to cause a massive outbreak in the Indian Ocean region [1], which subsequently spread to the Americas in 2013, resulting in millions of cases [2]. Ross River virus (RRV) causes thousands of cases a year in Australia [3] and has previously caused outbreaks in the Pacific Islands [4]. Mayaro virus (MAYV), traditionally confined to sylvatic transmission in South America, was recently isolated in Haiti for the first time, serving as a warning that urban transmission can occur [5]. Arthritogenic alphaviruses produce a disease that is characterized by fever, myalgia, and polyarthralgia, which can last for months or even years [6,7]. Recent CHIKV epidemics have been associated with an increased number of deaths, suggesting that severe disease may be more common than is typically thought [8–10]. While the rise in global travel, the spread of mosquito vectors, and climate change are likely involved in the expansion of these viruses, it is conceivable that changes in human health have also facilitated spread. Specifically, worldwide obesity has tripled since 1975, and the World Health Organization (WHO) currently estimates that 1.9 billion adults are overweight or obese [11]. Concurrently, roughly 10% of the adult population worldwide is considered underweight [12]. Over- and underweight, along with nutrient deficiencies, can be classified together as malnutrition and commonly occur in the same country or even within the same family [13].

Both obesity and undernutrition are known to alter immunity and can lead to an increased risk of infection from several pathogens (reviewed in [14]). Additionally, obesity and undernutrition have been shown to increase disease severity and decrease the protective immune response for influenza virus [15–18]. For arthropod-borne viruses (arboviruses), some epidemiological reports suggest that obesity is associated with a higher likelihood of being seropositive for CHIKV [19,20], Sindbis virus (SINV) [21], and dengue virus (DENV) [22]. In contrast, several reports suggest that undernourishment may result in a decreased risk of contracting dengue fever [22]. In addition to susceptibility to infection, obesity has been identified as a risk factor for greater disease severity following infection with CHIKV [23] and DENV [22]. Undernourishment, in contrast, has been associated with protection from severe DENV disease [24,25]. However, these data are not conclusive as other studies have found either no association with DENV disease severity [26,27] or an increased risk of severe disease [22]. Depending on how malnutrition was defined, infants in the same study were found to have both an increased or decreased risk for severe DENV disease [28]. While the epidemiological

data are quite clear that obesity is associated with more severe disease outcomes for various arboviruses, this connection is not so clear for undernutrition.

To our knowledge, no studies have directly assessed the role of nutrition in arbovirus infection using animal models. Given the worldwide prevalence of these forms of malnutrition and arbovirus disease, it is essential to understand how these comorbidities interact. We sought to determine what role both over- and undernutrition play in alphavirus disease, replication, transmission, and virus evolution in a mouse model. To accomplish this, we fed mice different diets; a high-fat overnutrition diet (HFD) with 45% fat; a low-protein undernutrition diet (5% protein); and a control diet with similar characteristics to the other diets except for macronutrient content. Following a sustained feeding period, we infected mice with either CHIKV, MAYV, or RRV and followed infection. We observed altered virus replication and transmission to mosquitoes in obese mice. Furthermore, we noted a reduction in genetic diversity in virus isolated from obese mice. Our findings illustrate the importance of nutrition on virus infection and suggest that nutritional status may be an underappreciated host factor in arbovirus disease, transmission, and evolution.

## Materials and methods

### Ethics statement

All animal work was performed in accordance with the Animal Committee regulations of the Institut Pasteur in Paris, France, in accordance with the 2010/63 EU directive adopted on 22 September 2010 by the European Parliament and the European Union Council. Mouse protocol 2013–0012 was approved by the Ethics Committee on Animal Experimentation at the Institut Pasteur.

### Cells and viruses

BHK-21 and Vero were obtained from the American Type Culture Collection and grown in DMEM with 5% fetal bovine serum (FBS) and 1% penicillin/streptomycin (P/S) at 37˚C with 5% CO2. Aag2 and U4.4 mosquito cell lines were initially obtained from G.P. Piljman, Wageningen University, the Netherlands and maintained at 28˚C with 5% CO2 in Schneider's medium supplemented with 10% FBS, 1% nonessential amino acids, and 1% P/S. An Asian genotype of chikungunya virus (CHIKV) was rescued from an infectious clone of a virus initially isolated in the Caribbean [29]. MAYV strain TRVL 4675 and RRV strain T48 were obtained from the Centers for Disease Control and Prevention (CDC, Fort Collins, Colorado, USA) and infectious clones were constructed under an SP6 promoter in the same plasmid backbone of the CHIKV clone described above. The MAYV infectious clone used in this study has been described previously [30]. The RRV infectious clone will be described further in a separate manuscript. A modified version of the MAYV plasmid that contained the nanoLuc (nLuc) reporter gene in the nsp3 protein was constructed as previously described [31]. Recovery of infectious virus from plasmid stocks was performed as previously described [32] except that Xfect RNA transfection reagent (Takara) was used to introduce RNA into cells. Virus titers were calculated by plaque assay as previously described [33].

### Mice

C57BL/6N (Black and albino) and Balb/c mice were obtained from Charles River at 4–6 weeks of age. For studies with CHIKV, male C57BL/6N were used. For all other studies, female mice were used. All biosafety level 3 (BSL3) work was performed in isolator units in accordance with Institut Pasteur regulations.

## Mouse Infections

Mice were infected intradermally via the hind left-footpad as previously described [34]. For experiments involving CHIKV and MAYV, $10^4$ PFU of virus in 50 μl was used for infections. For experiments involving RRV, $10^3$ PFU of virus in 50 μl was used for infections. All virus dilutions were performed in RPMI-1640 media with 10 mM HEPES and 1% FBS. Mice were weighed daily following infection and monitored for morbidity. Inflammation at the site of injection was assessed by measuring the width of the infected footpad using a digital caliper. Blood collections were performed using the submandibular route with a 5 mM lancet (Goldenrod) into a gold-top microtainer tube (BD, Franklin Lakes, NJ USA). Serum was separated via centrifugation at 5,000 x g for 10 minutes and transferred to a fresh tube before storage at -80˚C. For infections and blood collections, mice were anesthetized using either isoflurane inhalation or a ketamine/xylazine (10 and 1 mg/ml, respectively) mix introduced by intraperitoneal (IP) injection. Infection of RRV in Balb/c mice varied slightly. Mice were treated with 0.1 mg in 100 μL of IFNAR1 blocking antibody (MAR1-5A3, Santa Cruz Biotechnology, [35]) via the intraperitoneal route one day before infection to increase their susceptibility to infection. Viremia was measured in all studies by plaque assay unless otherwise noted.

## Histopathology and organ titration

Tissues were harvested from mice and fixed in 2% paraformaldehyde. The Virginia Tech Animal Laboratory Services (ViTALS) performed paraffin embedding, sectioning, and staining with hematoxylin and eosin, and two independent anatomic pathologists read the slides. For viral titers from organs, tissues were harvested aseptically into RPMI media with 10 mM HEPES and 1% FBS containing a sterile steel 5mm pellet to a final concentration of 10% weight by volume. The tissues were homogenized using a TissueLyser II (QIAGEN) for 2 minutes at 30 cycles per second. Plaque assays were performed on the cleared homogenate.

## Diets and feeding experiments

All diets used in this study were obtained from SAFE diets (Augy, France). These custom diets were designed to vary as little as possible from one another except for macronutrient ratios. The obesogenic diet was a slightly modified version of a standard high-fat diet that has been used previously [36]. Four-week-old mice were fed the diets for 8 and 10 weeks. S1 Table displays the macronutrient and mineral composition of each diet. Food and water were provided to mice *ad libitum*. Mice were weighed weekly or biweekly prior infection to monitor the effect of diet on weight. All groups of mice were maintained on the specified diet for the entire duration of the experiment, including after infection. Before infection, mice were fed for 8–10 weeks on the different diets; a 45% fat obesogenic diet, a 5% protein undernourishment diet, and a control diet. Throughout the rest of the manuscript, we will refer to mice fed these diets as obese, LP, and lean, respectively.

## ELISA

We quantified leptin levels in the serum of mice using the mouse/rat leptin Quantikine ELISA kit (R&D Systems) according to the manufacturer's instructions. The optical density values obtained were fit to a standard curve, and the data are expressed as ng/mL of leptin.

## Quantitative Reverse Transcription PCR (qRT-PCR) and Genome to PFU calculation

Genomic sense viral RNA was quantified using qRT-PCR. qRT-PCR was performed on unextracted serum samples that were first heat inactivated at 50˚C and then diluted 1:5 in phosphate-buffered saline (PBS). This method was shown to be more sensitive than when RNA extraction was included [37], a result which we confirmed here (S1 Fig). qRT-PCR was performed with the NEB Luna One-Step RT-qPCR for Probes mix (NEB, Ipswich, MA, USA). Primers and probes used in the study are listed in S2 Table and were synthesized by Integrated DNA Technologies (IDT, Leuven, Belgium). qRT-PCRs were performed according to the manufacturer's instructions except for several modifications. Trehalose (Sigma-Aldrich, St. Louis, MO, USA), betaine (Sigma-Aldrich), and bovine serum albumin (NEB) were added at a final concentration of 0.15 M, 0.5 M, and 1 mg/mL, respectively. Cycling conditions were as follows on a Bio-Rad CFX-96 (Hercules, CA, USA); 55˚C for 20 minutes for reverse transcription, 95˚C for 5 minutes for initial denaturation and polymerase activation followed by 40 cycles of 95˚C for 10 seconds (denaturation), and 60˚C for 1 minute (annealing/extension). Readings to measure amplification were taken after the combined annealing/extension step. The number of genomes/mL was calculated based on a series of dilutions of RNA that was generated by *in vitro* transcription from the full-length infectious clones mentioned previously to generate a standard curve. The Ct values of the samples were then fit to the standard curve to calculate the genomes/mL (GE/mL). The genome to PFU ratios (GE:PFU) was calculated by dividing the GE/mL by the PFU/mL.

## *In vivo* imaging

Albino C57BL/6 mice were used for *in vivo* imaging studies. Mice that were previously fed the different diets for 8-weeks were infected with $10^4$ PFU MAYV expressing nLuc in the hind left footpad. Following infection, mice were monitored for viremia and luminescence. Mice anesthetized with ketamine/xylazine were intraperitoneally injected with furimazine (20 μg/mouse, a gift of Yves Janin, Institut Pasteur) and imaged within 5 minutes of furimazine administration for 15 seconds. Photon flux (photons/second) was quantified using Living Image software (Perkin-Elmer). All images contained at least one uninfected control animal. For data analysis, we selected the hind left footpad as the region of interest (ROI). We then normalized the flux values within the ROI of each infected mouse to the flux of the ROI of the uninfected mouse present in the same image. These normalized data were then log-transformed to produce a normal distribution before performing statistical comparisons.

## Next-generation sequencing (NGS) library preparation

RNA was extracted from serum and cell culture supernatant samples using the ZR-96 Viral DNA/RNA Kit from Zymo Research (Irvine, CA, USA). Samples were then DNase treated with TURBO DNase (ThermoFisher) and purified using RNAClean XP beads (Agencourt). Library preparation for NGS was performed using the Trio RNA-Seq kit (NuGEN), which creates, amplifies, and fragments cDNA and then depletes ribosomal RNA. We then sequenced the libraries on an Illumina NextSeq500 using single-end 150bp reads.

## NGS data analysis

Generated sequencing files were first trimmed for low-quality bases and adapter sequences using BBDuk (BBMap—Bushnell B.—sourceforge.net/projects/bbmap/). To generate a consensus sequence for comparisons, we aligned the fastq files from the MAYV stocks that were

used for infection to a MAYV TRVL 4675 reference sequence (Accession MK070492.1) using BBMap. We next called variants using LoFreq and filtered the resulting vcf file to contain only variants above 50% [38]. Finally, we used vcftools to generate a new consensus fasta file based on the variants above 50% [39]. The trimmed reads were then used as an input for analysis using the ViVAN pipeline [40] with a coverage cutoff of >100x and a p-value filter of 0.05. The ViVAN outputs provide an array of genetic diversity indices, including Shannon entropy, region variation, and region heterogeneity. Analysis with Vivan was performed as previously described [29,40–42]. Briefly, for each position throughout the viral genome, base identity and their quality scores were collected. Each variant was determined to be valid using a generalized likelihood-ratio test (used to determine the total number of minority variants) and its allele rate was modified according to its covering read qualities based on a maximum likelihood estimation. Additionally, a confidence interval was calculated for each allele rate. In order to correct for multiple testing, the Benjamini-Hochberg false-discovery rate was set at 5%. Shannon entropy provides a measure of genetic diversity associated with the uncertainty of sampling a given genomic position; higher Shannon entropy indicates higher genetic diversity. The region variation is affected by selective pressures on the virus; a high variation rate suggests the accumulation of variant alleles which may be caused by positive selective pressures. Region heterogeneity, in contrast, measures replication fidelity; a virus with a more error-prone polymerase will have higher region heterogeneity. The equations for Shannon Entropy, region variation, and region heterogeneity are presented in our previous publications [40,41].

## Quantitative *in vivo* competition assay for empirical virus fitness measures

Relative fitness was measured using a qRT-PCR based direct competition assay previously described in [43] with some modifications. Serum was taken two days post-MAYV infection from mice fed either the control, 45% fat, or 5% protein diet to compete against a genetically marked reference virus with seven synonymous changes in the hypervariable portion of the nsp3 protein, which has previously been shown to tolerate modifications [44]. 30,000 PFU from each serum sample was mixed with 30,000 PFU (1:1 ratio) of the marked virus and then used to infect female 6-week old lean C57BL/6 mice in the hind left footpad. Serum was taken one and two days post-infection and left footpads were harvested following euthanasia on the second day post-infection.

## *In vivo* transmission experiments

*Aedes aegypti* female mosquitoes belonging to the 7th generation of a colony coming from wild mosquitoes collected in Kamphaeng Phet Province (Thailand) were used for these studies. The mosquitoes were reared at 28˚C, 70% relative humidity, under a 12 h light: 12 h dark cycle, and fed with 10% sucrose *ad libitum*. Groups of female mosquitoes were separated into cartons and eight days post-eclosion were allowed to feed on ketamine/xylazine anesthetized mice, which had been infected with MAYV 2 days previously, for 30–45 minutes. Fully engorged females were separated into new cartons and placed at 28˚C, 70% humidity, under a 12 h light: 12 h dark cycle with freely available sucrose. After ten days, the mosquitoes were anesthetized using Triethylamine (Sigma-Aldrich) and bodies and saliva were collected to assess infection and transmission, respectively. Bodies were homogenized using the TissueLyser II (QIAGEN) for 2 minutes at 30 cycles per second in RPMI containing 20% FBS, antibiotic/antimycotic mix, and 10 mM HEPES (herein called mosquito diluent). Saliva was collected by placing the proboscis in a barrier pipette containing 10μL mosquito diluent for 30 minutes. Following completion, the contents of the tip were pipetted into a tube containing 50μL mosquito diluent. The presence of virus was assessed by plaque assay and qRT-PCR.

## Statistics

All statistics were performed in GraphPad version 8. Weight and footpad swelling comparisons were performed using a two-way ANOVA with Tukey's multiple comparison test. Virus or RNA levels were compared using either a one-way or two-way ANOVA with Dunnett's correction for multiple comparisons. When analyses included a component of time, a repeated measures analysis was included since we followed the same individuals over time. Transmission/infection rates for individual mice and antibody titers were determined to follow a non-normal distribution; thus, we used the Kruskal-Wallis test with Dunn's correction for multiple comparisons. Total transmission and infection rates were compared using a two-tailed Fisher's exact test. Survival was compared using a Log-rank (Mantel-Cox) test. For comparisons between extracted and unextracted samples for qRT-PCR, data were compared using a two-tailed paired t-test. All experiments were repeated 2–3 times unless otherwise noted. The data are presented as the result of one representative experiment.

## Results

### The effect of diet on weight change and leptin levels in mice

Mice gained considerably more weight on the high-fat diets compared to the 5% protein or control diets (Fig 1A presents the change of weight as a percentage while Supp. Fig 2A–2C presents the weights in grams, $p < 0.0001$ at each time point after week 2). Undernourished mice experienced only slightly lower average weights than lean controls ($p < 0.05$ after eight weeks, except for Balb/c mice. S2C Fig). To further assess nutritional status, we measured serum levels of the adipokine leptin. Leptin, a pro-inflammatory adipokine primarily secreted by adipocytes, is a blood marker that generally increases during obesity and decreases in severe undernourishment [45]. Obese mice had significantly increased levels of leptin in their serum following 8–10 weeks of feeding compared to lean controls (Fig 1B, $p < 0.0001$). Leptin levels in LP mice were not significantly different from controls ($p = 0.2935$).

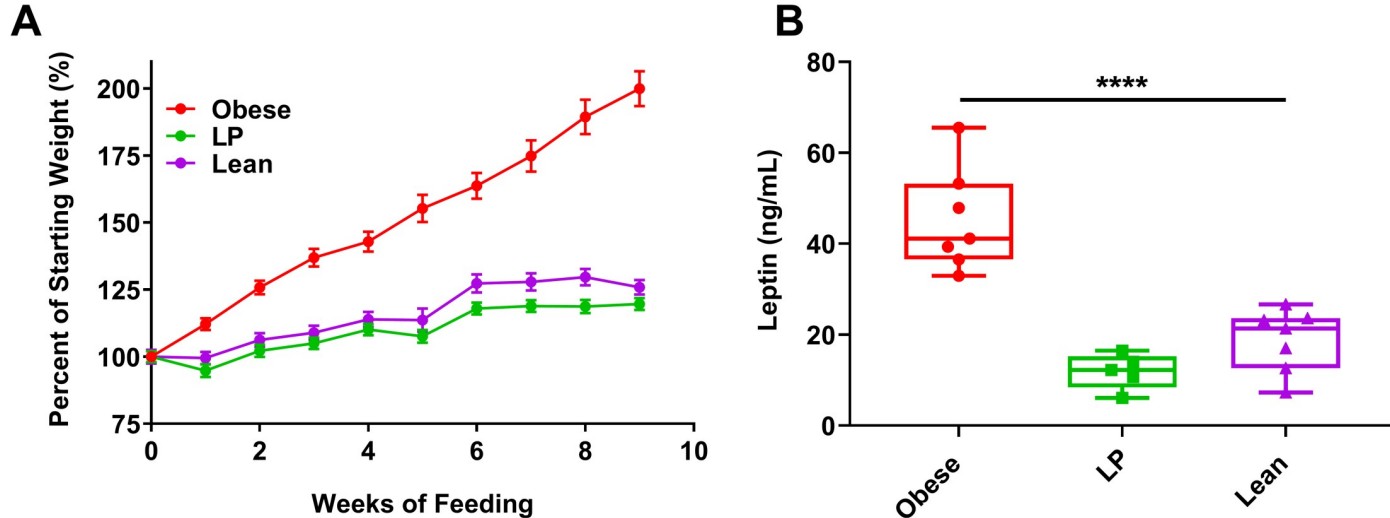

**Fig 1. Weights of mice following feeding on different diets.** Groups of C57BL/6N mice (7 mice per group) were fed a 45% fat (Obese), 5% protein (LP), or a control (Lean) diet for 8–10 weeks. **A.** Weights were measured weekly or bi-weekly following the initiation of feeding. Weights are plotted as the percent change compared to initial weight at the onset of feeding. **B.** Leptin was measured following 8–10 weeks of feeding using an ELISA. Statistical comparisons were made using a one-way ANOVA with Dunnett's correction compared to the control group. The level of significance is represented as follows—**** $p < 0.0001$. For A, the error bars represent standard error. For B, the whiskers represent the minimum and maximum points, the box limits represent the 25th and 75th percentile, and the horizontal line represents the median. Each graph represents data obtained from at least two independent experiments. The results from one representative experiment are presented.

## Obese mice experience more severe morbidity following alphavirus infection

Our goal was to determine the effect of nutrition on disease burden upon infection with an arthritogenic alphavirus. To this end, we infected mice with MAYV following the 8–10 week feeding period and monitored the mice for signs of morbidity, specifically weight change or footpad swelling, over time. Obese mice experienced significantly greater weight loss following MAYV challenge compared to lean mice (Fig 2A, p<0.05 for days 1, 6, and 10 post-infection, compared to lean controls. S3A Fig presents the data in grams). Obese mice also displayed significantly higher footpad swelling following MAYV infection compared to lean controls (Fig 2B, p = 0.0003 on day 2 post-infection compared to lean controls). Lean controls and LP mice displayed similar results following MAYV infection.

We next sought to determine if the differences in morbidity observed following MAYV infection were true for another arthritogenic alphavirus, namely CHIKV. Obese mice lost significantly more weight following CHIKV challenge compared to lean mice (Fig 2C, two-way ANOVA, all p<0.05 between days 5–7 compared to lean controls. S3B Fig presents the data in grams). No differences were observed between the LP and lean groups infected with CHIKV. Both obese and LP mice experienced increased footpad swelling compared to lean controls (Fig 2D, obese vs. lean p<0.05 on days 7–9 and LP vs. lean p<0.05 on days 7, 9, 10, and 11). However, despite the differences in footpad swelling, we did not observe differences in histopathology between any of the groups following CHIKV infection (S4 Fig). Furthermore, we did not observe differences in histopathology in the liver, spleen, heart, or brain. We observed infiltration of histiocytes in all the infected footpads, and overall scores of pathology were similar among the groups (S5 Fig).

To determine whether these results were specific to C57BL/6 mice, and could be further expanded to another arthritogenic alphavirus, we next infected adult Balb/c mice with RRV. Since previous reports found that these mice are refractory to RRV infection [46], we treated the mice with an anti-IFNAR antibody (MAR1-5A3) to block type-1 interferon signaling one day before infection, as this is known to increase susceptibility to viral infection [35]. Both obese and LP mice experienced significantly increased weight loss after RRV challenge compared to lean control mice (Fig 2E, p<0.05 for both groups as compared to control on days 1–5 post-infection. S3C Fig presents the data in grams.). After infection, 2 and 3 mice from the obese and LP groups succumbed to infection, respectively, while none succumbed in the lean group (Fig 2F). Statistical comparisons for survival approached but did not reach, statistical significance (p = 0.138 and 0.055 for obese and LP, respectively, compared to control).

## Nutrition alters alphavirus replication kinetics

To evaluate the effect of diet on virus replication in mice, we infected mice with either MAYV, CHIKV, or RRV and then measured serum levels of virus over time either by qRT-PCR or plaque assay. On Day 1 post-MAYV infection, we observed increased levels of viremia and RNAemia for obese and LP mice compared to lean controls (Fig 3A and 3B; for viremia; obese vs. control—p = 0.0002 and LP vs. control—0.0128; for RNAemia; obese vs. control—p = 0.0008 and LP vs. control—0.0350). Obese mice displayed lower levels of both infectious virus and viral RNA 3 days post-MAYV infection compared to lean controls (viremia; p = 0.0293 and RNAemia; p = 0.0007). No differences were observed in viral RNA levels in the blood of mice infected with CHIKV except for day 3 post-infection where obese mice had significantly lower RNAemia than lean controls (Fig 3C, p<0.0001). There were no differences in replication at any time post-infection following RRV infection (Fig 3D). Given the similarities observed

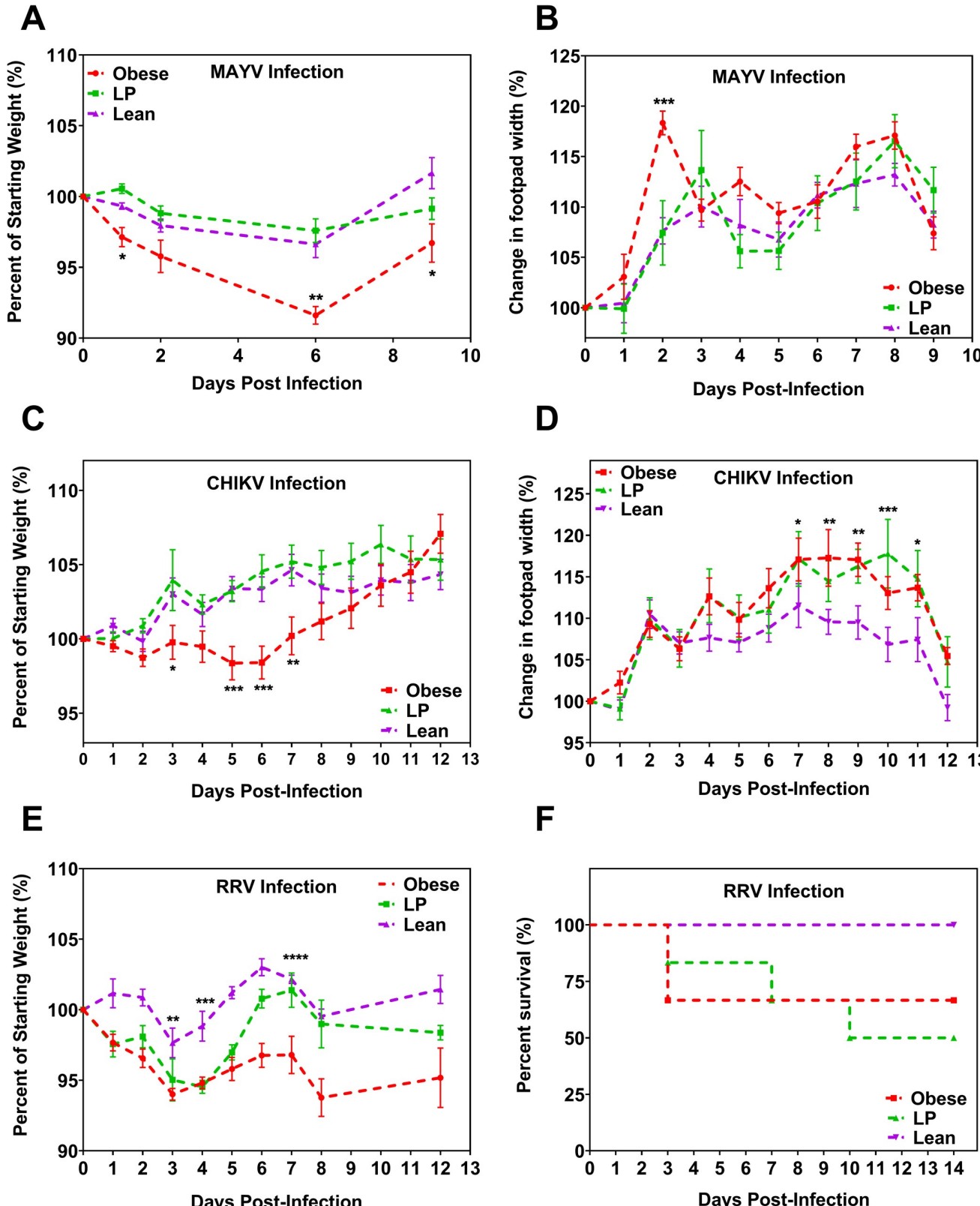

**Fig 2. Mice with differing nutritional status have altered morbidity following alphavirus infection.** Groups of C57BL/6N mice (A-D) or Balb/c (E-F) were fed either a 45% fat (Obese), 5% protein (LP), or control (Lean) diet for 8–10 weeks. Mice were then infected with $10^4$ PFU of either Mayaro virus

(MAYV, 7 mice per group) or chikungunya virus (CHIKV, 14 mice per group) or $10^3$ PFU of Ross River virus (RRV, 5 mice per group) in the hind left footpad and monitored for weight change (A, C, and E), footpad swelling (B, D), and, for RRV, mortality (F). For A-E, statistical analyses were performed by two-way ANOVA with Dunnett's correction compared to the control group. For F, statistical analysis was performed by the Mantel-Cox test. The error bars represent standard error. The level of significance is represented as follows—* $p < 0.05$, ** $p < 0.01$, *** $p < 0.001$, ****$p < 0.0001$. Besides E-F, each graph represents data obtained from at least two independent experiments. The results from one representative experiment are presented.

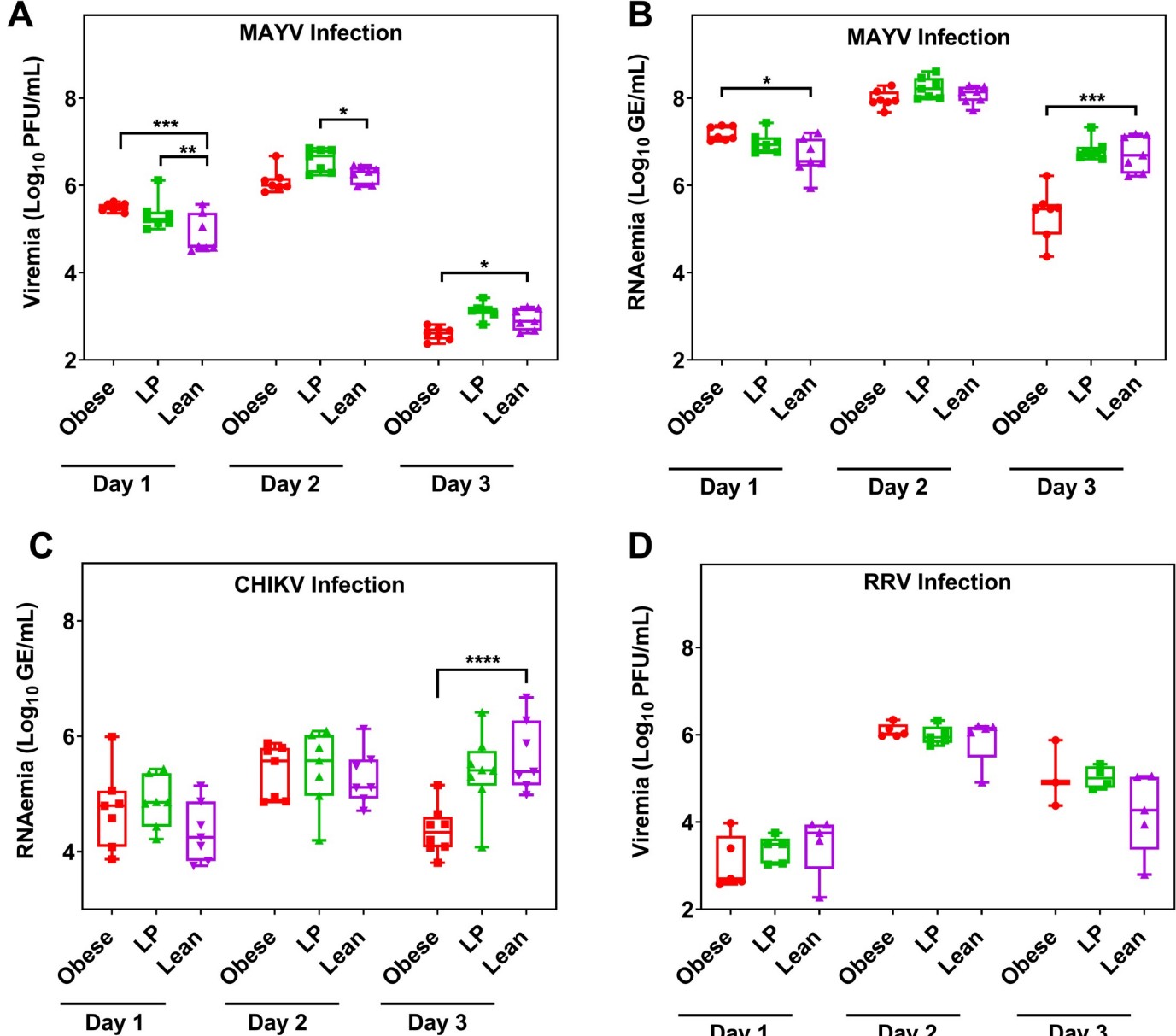

**Fig 3. Nutrition alters viral replication kinetics.** Groups of C57BL/6N (A-C, 7 mice per group) or Balb/c mice (D, 5 mice per group) were fed either a 45% fat (Obese), 5% protein (LP), or control (Lean) diet for 8–10 weeks. Mice were then infected with either Mayaro virus (MAYV, A-B), chikungunya virus (CHIKV, C), or Ross River virus (RRV, D). At different times post-infection, the virus was quantified either by plaque assay (A, D) or qRT-PCR (B, C). GE = genome equivalents. Statistical analyses were performed by two-way ANOVA with Dunnett's correction compared to the control group. The level of significance is represented as follows—* $p < 0.05$, ** $p < 0.01$, *** $p < 0.001$, **** $p < 0.0001$. The whiskers represent the minimum and maximum points, the box limits represent the 25th and 75th percentile, and the horizontal line represents the median. Except for D, each graph represents data obtained from at least two independent experiments. The results from one representative experiment are presented.

between CHIKV, MAYV, and RRV replication and pathogenesis in mice, we proceeded only with MAYV for all future studies.

In order to determine the effect of diet on viral dissemination and replication in the organs, we collected the spleen and liver from mice infected with MAYV 2 days post-infection and measured viral titers by plaque assay. We did not observe significant differences in viral titers between any group for either liver (S6A Fig) or spleen (S6B Fig).

## Obese mice infected with luciferase-expressing MAYV have increased local replication

To visualize local replication at the site of infection, we infected mice fed different diets with a MAYV expressing the nano luciferase protein (nLuc). Obese mice displayed a significant increase in luminescence in their hind-left footpads 6 days after infection compared to lean controls (Fig 4A, p = .0022). We did not observe differences in luminescence between the groups on day 2 or day 13 post-infection. Furthermore, at 15 seconds of exposure time on day 6 post-infection, nLuc expression was observed in the bodies of obese mice but not in the other groups. Representative images of each group on day 6 post-infection are presented for the obese (Fig 4B), LP (Fig 4C), and lean groups (Fig 4D).

## Virus isolated from mice fed a high-fat diet has reduced genetic diversity

To examine the role of diet on the viral genetic diversity, we performed NGS on the viral RNA isolated from mice that were fed different diets and then infected with MAYV. We sequenced viral RNA from serum samples that were isolated 2 days post-infection. Since there were no significant differences in the concentration of genomes in these samples between the groups (Fig 3B), we used the same volume for each sample as input for library preparation. Following Illumina sequencing, we used the ViVan pipeline [40] to generate diversity statistics to assess differences between viral genetic diversity in the virus from each group. Average sequencing coverage across the genome was the same for each group (S7 Fig). Overall and non-synonymous Shannon entropy, markers of genetic diversity, were significantly reduced in virus isolated from obese mice compared to lean mice (Fig 5A and 5B, p = 0.0082 and p = 0.0086, respectively). The difference between LP and lean mice approached statistical significance (p = 0.0587 and 0.0797 for total genome and non-synonymous Shannon entropy, respectively). We next compared region variation, a marker that is affected by selective pressures. Viral RNA from obese mice showed significantly lower region variation for both overall (Fig 5C, p = 0.0107) and non-synonymous (Fig 5D, p = 0.0129) sites compared to lean controls. We observed a trend towards lower region variation for both total and non-synonymous regions when we compared the LP and lean groups (p = 0.1044 and p = .1068, respectively). We also assessed region heterogeneity, which is not impacted by the accumulation of specific variations and is, therefore, a measure of replication fidelity. No significant differences were observed between any group for region heterogeneity (Fig 5E and 5F).

## Virus isolated from obese mice has reduced fitness in mice

We made viral fitness comparisons using both indirect measurements, genome to PFU ratio (GE:PFU), and direct measurements, competitive fitness assays. We observed no differences in GE:PFU among the groups on either day 1 or 2 post-infection (Fig 6A). However, we noted a significant increase in GE:PFU ratio between days 1 and 2 for the obese group, while the other two groups remained consistent (p = 0.0169 for day 1 vs. day 2 for the obese group, p>0.9999 for day 1 vs. day 2 comparison for the other two groups). Because viral fitness is dependent on the replication environment, we next sought to determine fitness differences of viruses isolated

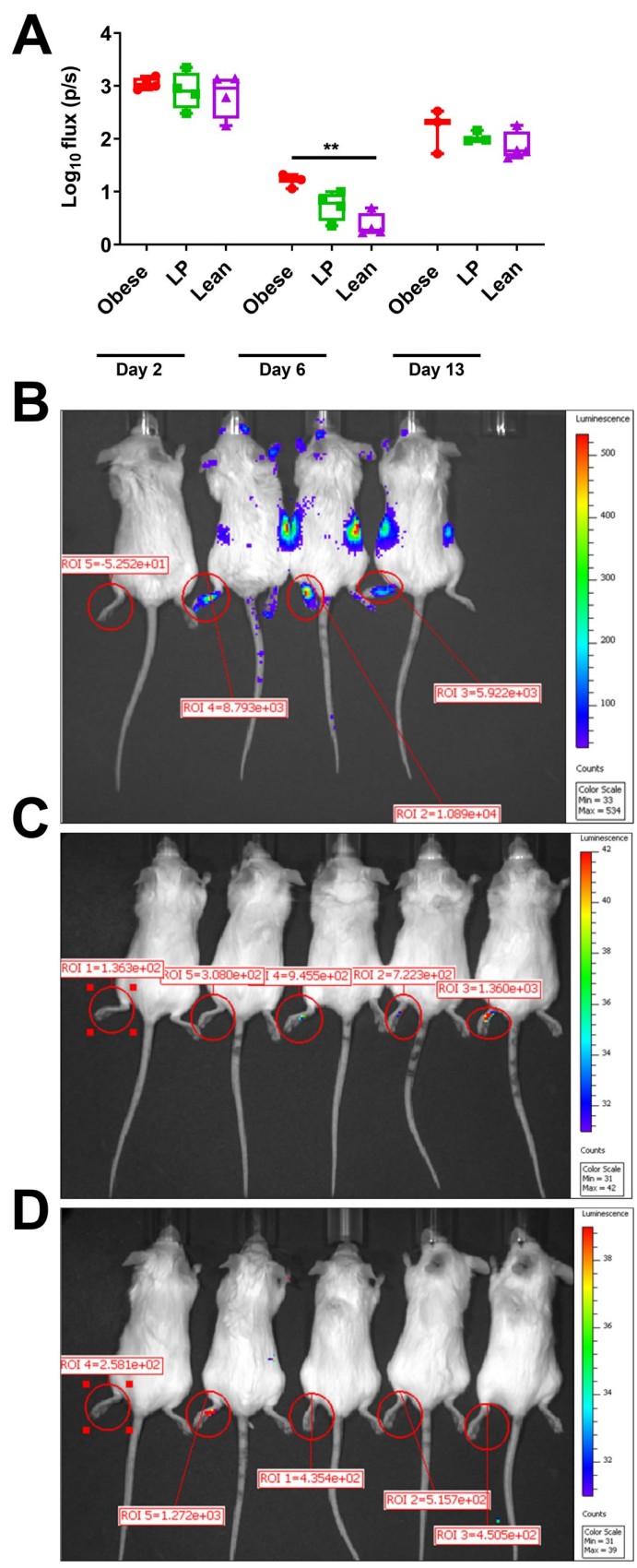

**Fig 4. Local viral replication is increased in obese mice.** Groups of albino C57BL/6N mice (5 mice per group) were fed either a 45% fat (Obese, B), 5% protein (LP, C) or control diet (Lean, D) for 8–10 weeks. Mice were then infected with Mayaro virus (MAYV) expressing nano-luciferase (nLuc) to track replication. At different time points post-infection, mice were anesthetized, injected intraperitoneally with 20 μg/mouse of the furimazine substrate and imaged using a Perkins Elmer IVIS Spectrum machine. Mice were imaged for 15 seconds within 5 minutes of substrate administration, and photon flux (photons/second) was quantified in the hind left footpad using Living Image software (A). The whiskers represent the minimum and maximum points, the box limits represent the 25th and 75th percentile, and the horizontal line represents the median. The data are presented as flux or photons/second (p/s) present within the left footpad. The flux is used to quantify the amount of light emitted from the footpad. Representative images of mice fed different diets, infected with MAYV-nLuc, and then imaged six days following infection (B-D). An uninfected control mouse is presented as the leftmost mouse in each image. Statistical analyses were performed by two-way ANOVA with Dunnett's correction compared to the control group. The level of significance is represented as follows —** p<0.01.

from mice fed on different diets when faced with new replication conditions, naïve lean mice. To assess fitness, we mixed a marked reference virus (REF) at a 1:1 ratio (PFU:PFU) with virus from serum isolated 2 days post-infection with MAYV. The marked reference virus contains seven synonymous mutations introduced in the hypervariable region of the nsp3 gene, which facilitates differentiation from the wild-type (WT) sequence. Fitness was measured using a Taqman qRT-PCR based assay that distinguishes between WT and REF sequences (S8 Fig). We then used the 1:1 mix to infect new lean mice in the hind left footpad. Virus derived from obese mice had a significant reduction in fitness at the injection site compared to lean controls (Fig 6B, p = 0.0097). Despite the early difference in fitness, virus derived from obese mice was able to recover fitness, as no differences were observed on day 1 or 2 post-infection in the serum among any group (Fig 6C and 6D).

## *Aedes aegypti* mosquitoes have lower vector competence for virus derived from obese mice

Since most arboviruses must efficiently infect mosquitoes to sustain transmission, we next assessed whether mosquitoes that fed on viremic mice fed different diets would have altered vector competence. We infected groups of C57BL/6 mice fed different diets with MAYV and, two days later, groups of *Ae. aegypti* mosquitoes (n = 16–20) were allowed to feed on individual mice (n = 5), resulting in 96 blood-fed mosquitoes per group. The mosquitoes were held for 10 days before bodies, to assess infection, and saliva, to assess transmission potential, were collected. The amount of infectious virus present was the same in the serum of mice from each group on the day of the transmission (Fig 7A, p = 0.3722 and p = .9863 for the obese and LP groups compared to lean controls, respectively).

To assess the infection rate of the mosquitoes fed on individual mice, we first analyzed the data by measuring infection and transmission rates from each mouse. Mosquitoes that fed on obese mice had reduced infection and transmission rates compared to lean controls (Fig 7B and 7C, p = 0.0451 and 0.0443, respectively). The infection and transmission rates observed in LP and control groups were not significantly different. The infection rates for obese mice varied from 6.3–55%, LP from 15–50%, and lean controls from 35–75%. The transmission rates for obese mice varied from 0–5%, LP from 0–20%, and lean controls from 5–15%. 60% and 20% of obese and LP mice had no mosquitoes found to be transmission positive following feeding, whereas all the lean controls resulted in transmission positive mosquitoes.

When we combined all mosquitoes for all mice in each group, we found that infection rates were significantly lower for mosquitoes fed on both obese and LP mice compared to lean control (Fig 7D, p = 0.0001 and 0.0005, respectively). Only 2% of all mosquitoes exposed to obese mice contained infectious virus in their saliva. In contrast, 7.3% of LP mice and 8.3% of lean mice contained infectious virus in their saliva. However, the difference between obese and lean

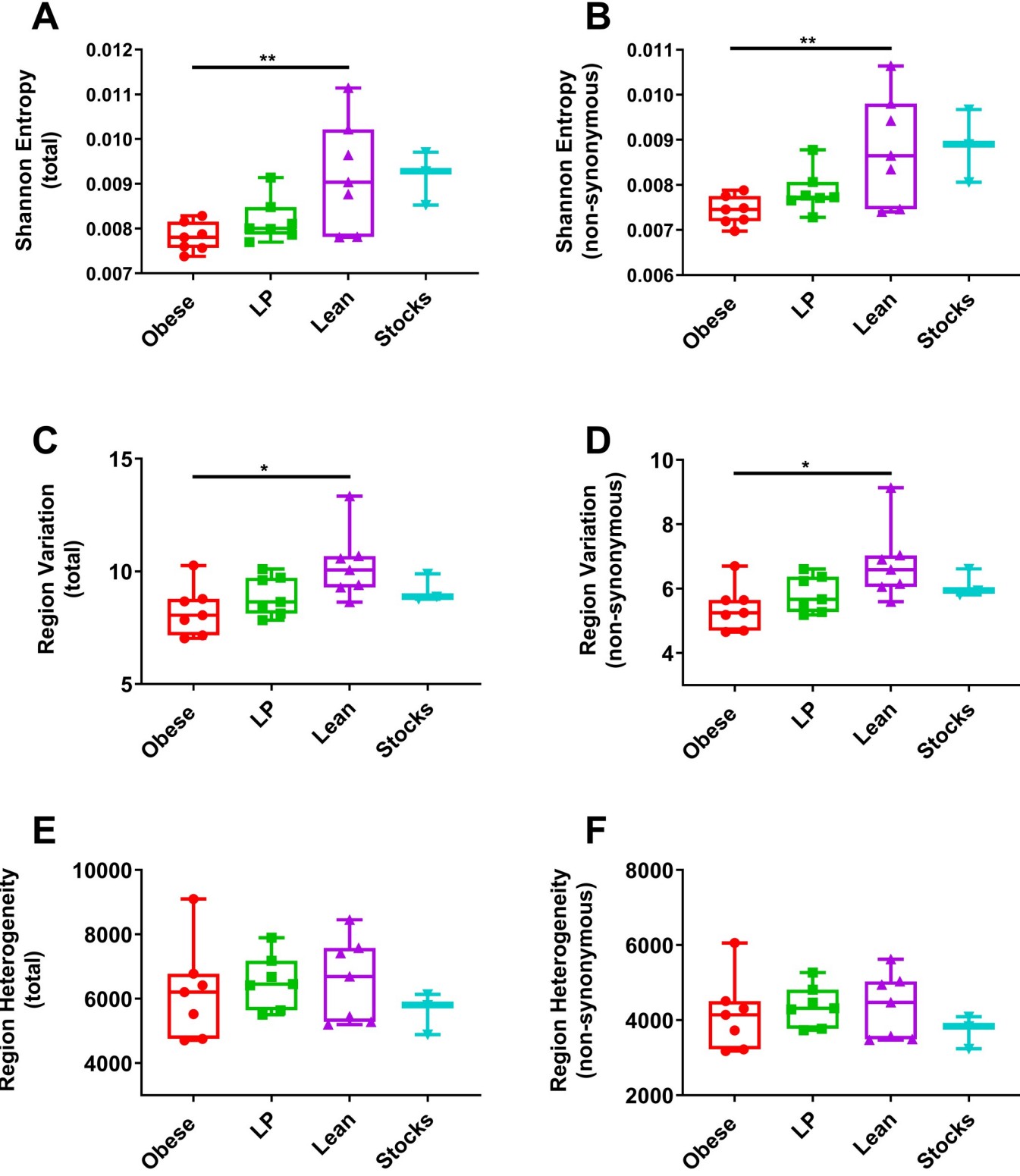

**Fig 5. Viral replication in obese mice results in decreased genetic diversity.** Groups of C57BL/6N mice (7 mice per group) were fed either a 45% fat (Obese), 5% protein (LP), or control diet (Lean) for 8–10 weeks. Mice were then infected with Mayaro virus (MAYV) and blood was collected two days later. RNA was isolated from the serum and next-generation sequencing libraries were prepared and then sequenced on an Illumina NextSeq 500. The FASTQ files were trimmed for adapter and low-quality sequences using BBDuk. The trimmed reads were then analyzed using the ViVAN pipeline with a coverage cutoff of >100x and a p-value filter of 0.05. From the output of ViVAN, we compared both total and non-synonymous Shannon entropy (A-B), Region variation (C-D),

and Region heterogeneity (E-F) as markers of genetic diversity. The whiskers represent the minimum and maximum points, the box limits represent the 25th and 75th percentile, and the horizontal line represents the median. Statistical analyses were performed by one-way ANOVA with Dunnett's correction compared to the control group. The level of significance is represented as follows—* $p < 0.05$, ** $p < 0.01$.

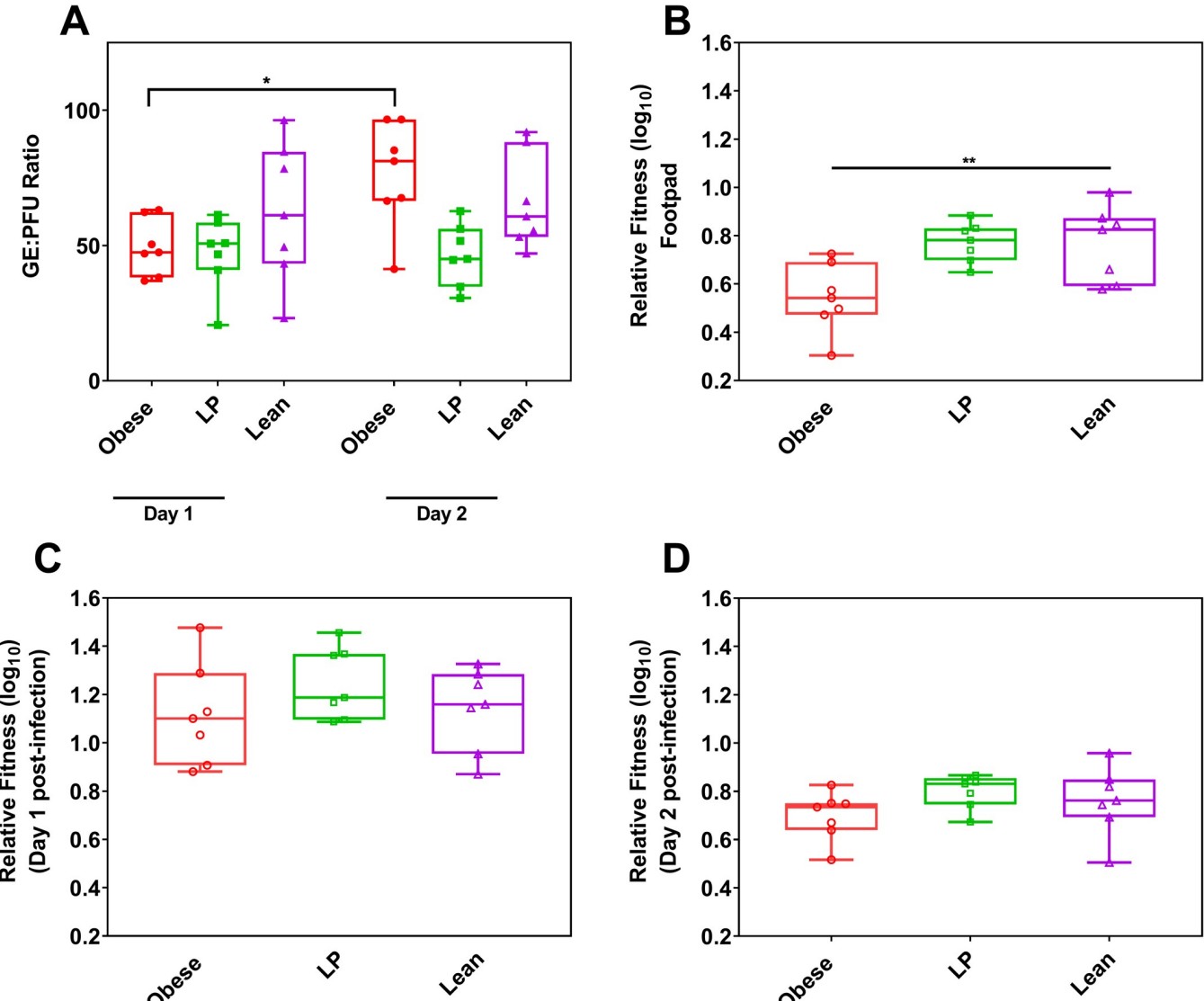

**Fig 6. Virus isolated from obese mice reduced early *in vivo* fitness.** Relative fitness of Mayaro virus (MAYV) isolated from mice fed different diets was tested by measuring an indirect measurement, genome to PFU ratio (GE:PFU), and an *in vivo* direct competition assay. Groups of C57BL/6N mice (5 mice per group) were fed either a 45% fat (Obese), 5% protein (LP), or control diet (Lean) for 8–10 weeks. Mice were then infected with Mayaro virus (MAYV) and bled at different time points post-infection. GE:PFU ratios were calculated by dividing the number of RNA genomes present in the serum of mice by the number of PFUs in the same sample (A). Statistical comparisons were performed using a two-way ANOVA with Dunnett's correction compared to the control group. For direct *in vivo* competition assays, serum containing MAYV collected on two days post-infection of individual mice fed different diets were mixed at a 1:1 ratio with a genetically marked reference virus and then this mixture was used to infect new, lean C57BL/6N mice. For each mouse tested, one new mouse was infected (n = 7). Following infection of mice with the mixtures, serum samples were collected for two consecutive days, the mice were euthanized, and footpads were collected. Fitness was measured using a multiplex qRT-PCR developed in-house to discriminate between the wild-type MAYV sequence and the marked reference virus. Fitness is shown for each group in the footpad (B) and serum for one (C) and two-days (D) post-infection. The whiskers represent the minimum and maximum points, the box limits represent the 25th and 75th percentile, and the horizontal line represents the median. Statistical analyses were performed by one-way ANOVA with Dunnett's correction compared to the control group. The level of significance is represented as follows—* $p < 0.05$, ** $p < 0.01$.

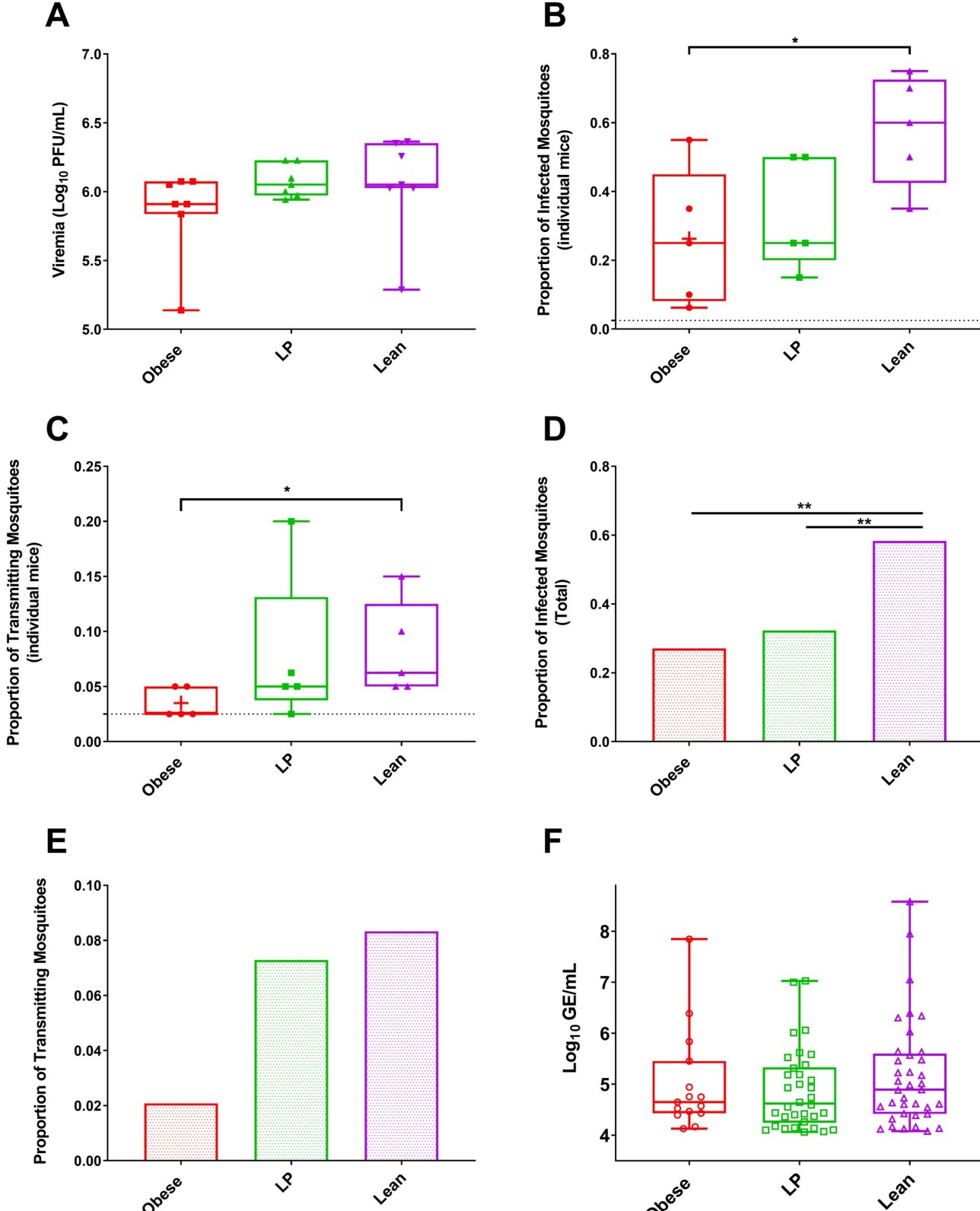

**Fig 7. Virus from obese mice decreased infectivity to mosquitoes and lower transmission rates.** Groups of C57BL/6N mice (7 mice per group) were fed either a 45% fat (Obese), 5% protein (LP), or control (Lean) diet for 8–10 weeks. Mice were then infected with Mayaro virus (MAYV), and two days

later were individually used to feed groups of *Aedes aegypti* mosquitoes. Serum was also collected to measure viremia two days post-infection by plaque assay (A). Mosquitoes were dissected ten days post-feeding on the viremic mice, at which point bodies (for infection, B and D) and saliva (for transmission potential, C and E-F) were collected for individual mosquitoes. The infection and transmission rates were calculated for the mice individually within a group (B-C) and for all individual mice combined within a group (D-E) based on the presence or absence of infectious virus tested by plaque assay. The number of RNA copies were calculated by qRT-PCR (F). The whiskers represent the minimum and maximum points, the box limits represent the $25^{th}$ and $75^{th}$ percentile, and the horizontal line represents the median. Statistical analyses for A-C and F were performed by one-way ANOVA with Dunnett's correction compared to the control group. For comparing rates in D and E, a two-tailed Fisher's exact test was used. The level of significance is represented as follows—* $p<0.05$, ** $p<0.01$.

mice did not reach significance for transmission in all mosquitoes (Fig 7E, p = .10). Finally, we compared the number of viral genomes in the saliva of mosquitoes fed on each group (Fig 7F), which is a proxy for the ability of an individual mosquito to transmit to a new host. No significant differences were observed between any of the groups.

## Obese mice have reduced chronic RNA levels

Arthritogenic alphaviruses can cause chronic infection, and viral RNA has been detected up to 18 months post-infection in humans [47]. Therefore, we next assessed whether nutrition affects viral RNA persistence in mice. We collected footpads from obese, LP, and lean mice 65 days post MAYV (Fig 8) or RRV (S9 Fig) infection to test for viral RNA levels. Obese mice had significantly lower levels of MAYV RNA in their footpads compared to lean mice (p = 0.0476). Obese mice also had lower levels of RRV RNA in their footpads 65 days post-infection compared to controls (p = 0.0470). No infectious virus was observed in any of these samples.

## Discussion

Obesity is a global issue, with rates tripling since 1975 [12]. At the same time, roughly 10% of adults globally are underweight [12]. Obesity has been associated with increased disease severity upon infection with CHIKV in humans [17,23,48]. In contrast, no information exists regarding the relationship between alphavirus infection/disease and undernutrition. In this study, we sought to clarify the relationship between nutrition and alphavirus infection. More explicitly, we aimed to examine the role of both obesity and undernutrition on infection, transmission, and evolution of arthritogenic alphaviruses, focusing on MAYV but including other important human pathogens, CHIKV and RRV. We fed mice different diets leading to either obesity or undernutrition and compared them to lean control mice. Obese mice experienced more severe morbidity upon infection for all three arthritogenic viruses tested. These results are consistent with previous studies showing an association between body-mass index (BMI) and more severe disease sequelae in humans following CHIKV infection [23]. In these studies, undernutrition did not result in more severe morbidity in acute arthritis models of infection with either CHIKV or MAYV. Our use of only one type of undernutrition, protein-energy malnutrition, limits our ability to generate definite conclusions on undernutrition generally since undernutrition also includes micronutrient deficiencies or general energy deficits. It is possible that other forms of undernutrition alter infection differently than protein-energy malnutrition. To our knowledge, no studies have specifically looked at the relationship between undernutrition and alphavirus disease severity. However, several conflicting studies with dengue virus (DENV) have shown that undernutrition may be associated with both protection [24,25] or increased susceptibility to [22] DENV disease.

We next showed that obesity alters viral replication. For MAYV and CHIKV infection, we observed increased early viral replication in the blood of obese mice compared to lean controls. It is unlikely that this was due to faster local replication at the injection site (hind left footpad) since we observed no differences in viral replication on day 2 post-infection in obese mice

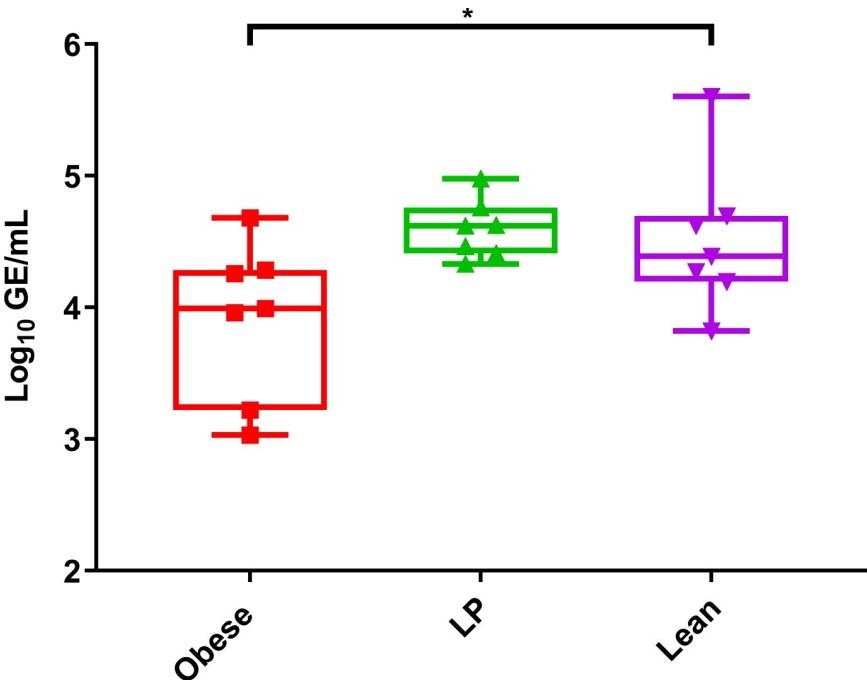

**Fig 8. Obese mice have reduced chronic RNA levels.** Groups of C57BL/6N mice (7 mice per group) were fed either a 45% fat (Obese), 5% protein (LP), or control (Lean) diet for 8–10 weeks. Mice were then infected with Mayaro virus (MAYV) and after 65 days post-infection footpads were collected. Viral RNA was quantified using qRT-PCR. The whiskers represent the minimum and maximum points, the box limits represent the 25th and 75th percentile, and the horizontal line represents the median. Statistical analyses were performed by one-way ANOVA with Dunnett's correction compared to the control group. The level of significance is represented as follows—* $p < 0.05$.

footpads (Fig 4A). These results suggest that diet did not alter early replication in the joint. Given the inflammatory nature of obesity (reviewed in [49]) and the fact that obesity alters epithelial cell permeability [50], it is possible that the difference in early virus replication may reflect faster escape from the tissues, resulting in a more rapid systemic spread. No clear, repeatable differences in viral replication were observed between LP mice and lean controls. It is possible that the diet used in these studies was not sufficiently low in protein and, as such, the mice were not fully malnourished. We used a 5% protein diet for these studies. Previous studies using a 2% protein diet found increased viral replication with both influenza virus [15] and Sendai virus [51]. However, this was not the case with respiratory syncytial virus, as no differences in viral replication were observed in 2% protein fed mice compared to lean controls [52]. According to Abrahams et al. [53], the range of energy from protein in forty Sub-Saharan African countries as a percentage is 6.08–13.47 ± 1.6, suggesting our use of 5% was more reflective of the current reality than 2%.

We next assessed the genetic diversity of virus isolated from mice fed different diets. Virus isolated from obese mice had both significantly lower total and non-synonymous Shannon entropy, a measure of information content that is maximum when all alleles are equifrequent, compared to lean mice. Furthermore, obese mice displayed a significantly lower variation rate, which identifies the specific accumulation of mutations in a population that may be associated with selective pressures, than lean controls. The difference between virus isolated from undernourished and lean mice approached but did not meet statistical significance for these metrics. Finally, we assessed region heterogeneity, a marker of replication fidelity not associated with selection, and found no differences among any of the groups. Collectively, these observations

led us to hypothesize that there may be some selective pressure in obese mice that reduces viral genetic diversity. A previous study with Sendai virus in mice fed different diets found no differences in genetic diversity in virus isolated from bronchiolar lavage fluid [54]. It is possible that differences between Sendai virus, which is negative-sense, and MAYV, which is positive sense, account for this. For example, positive sense viruses have greater variability in mutation rates than negative sense counterparts [55]. Furthermore, it has been suggested that codon usage in positive sense viruses matches their hosts more closely than negative sense viruses, possibly because the genome acts as an mRNA [56]. Therefore, evolution likely differs significantly between negative and positive sense RNA viruses.

Since we sequenced virus from the blood from mice infected in the footpad, it is also conceivable that bottlenecks within the mice accounted for the observed differences. It is plausible that the virus replicated in the blood of obese mice more slowly later in infection due to the host immune response or other factors, which could account for both the reduction in viral titer on day 3 post-infection and the reduced genetic diversity. Obesity is known to alter the relative amounts of different leukocytes, potentially altering the availability of susceptible cells during infection or leading to a skewed cytokine response [57,58]. It is possible that differences in the immune status of mice from each group result in different selective pressures on the virus population. Several studies have shown that the innate immune system is involved in viral evolution [59,60]. Finally, obese mice are expected to have more adipocytes (i.e., fat cells) while undernourished mice are expected to have fewer. Adipocytes secrete a variety of cytokine-like molecules called adipokines that alter immune response and energy utilization [61,62]. Furthermore, adipocytes differentiate from fibroblasts [63], which are known targets of alphavirus replication [64,65], potentially making them targets for alphavirus replication. Future studies will aim to uncover the mechanism behind the differences in viral genetic diversity observed in mice fed different diets.

Previous studies from our group have shown that sufficient genetic diversity and replication fitness is required to maintain adaptability, dissemination, and efficient transmission [41,66,67]. Therefore, we hypothesized that virus isolated from obese mice might have reduced fitness and adaptability in a new environment. To test this, we first individually mixed MAYV isolated from the serum of mice fed different diets with a genetically marked reference virus and exposed them to a new environment by infecting new lean mice with this mix. Virus isolated from obese mice was found to have reduced fitness at the site of injection in the new environment compared to virus derived from lean mice. However, viral fitness was able to recover in the serum, as we observed no differences on days one or two post-infection. Future studies should probe fitness changes after multiple passages in hosts with different nutritional status, which might occur during an actual outbreak.

We next performed a mosquito transmission experiment by exposing mosquitoes to viremic mice previously fed different diets. For arboviruses, most transmission cycles require both the vertebrate and invertebrate host, which are highly divergent environments, a fact that restricts arbovirus evolution [68,69]. Mosquitoes that fed on obese mice displayed reduced infection and transmission rates compared to mosquitoes fed on lean control mice, despite no differences in viremia at the time of bloodmeal acquisition. These results point towards the decreased adaptability of virus isolated from obese mice, possibly due to the reduced genetic diversity observed in sequencing data. It is also possible that lipids or other factors in the blood that are altered in obesity or lipid changes in the virions themselves altered rates of transmission to mosquitoes in obese mice. Only mosquitoes that were fully bloodfed were selected for testing, and viremia levels were not different between the groups, so each mosquito likely received roughly the same amount of infectious virus. These data suggest that obese individuals may provide a certain level of refractoriness against subsequent viral transmission.

Nutrition has been linked previously to virus fitness in mice fed either a selenium or vitamin E deficient diet and then infected with Coxsackie B3 virus; the virus was found to evolve quickly to a more virulent genotype [70,71]. Nelson et al. observed similar results in selenium-deficient mice infected with influenza virus, as a new virulent strain emerged from a previously low virulence progenitor strain in selenium-deficient but not selenium-adequate mice [72]. Finally, 10 passages of Sendai virus in mice fed a control, high-fat, or low-selenium diet did not result in differences in genetic diversity (as measured by Shannon entropy) in the virus population. However, the virus populations displayed phenotypic differences upon infection of a new host, suggesting that nutritional status influenced the virus population and resulting transmission. It is possible that their sequencing approach was not sensitive enough to detect genetic differences within the population. Given the prevalence of obesity, undernutrition, and nutrient deficiencies, future studies should elucidate the influence of nutritional status on virus fitness and adaptation to novel environments.

Arthritogenic alphaviruses are known to cause chronic inflammation, and CHIKV RNA and protein have been found in an infected human up to 18-months post initial infection [47]. Furthermore, CHIKV has been found in the infected joints of mice up to 16 weeks post-infection [73]. Therefore, we sought to assess whether nutrition influences persistent alphavirus replication in the joints of mice. We found that obese mice had significantly lower levels of RNA for both MAYV and RRV 65 days post-infection compared to lean controls. However, it is unclear what role persistent viral replication plays in chronic alphavirus disease, as long-term disease in humans is strongly associated with the presence of pro-inflammatory cytokines but not necessarily viral RNA [74]. Therefore, obesity may be involved in persistent alphavirus replication, which may be involved in chronic disease.

These studies highlight the importance of nutrition in alphavirus infection and provide further support for the need to consider this population in epidemiological studies and vaccine trials. Furthermore, malnutrition represents a simple way to alter the immune landscape of an animal, making it possible to identify host factors involved in virus replication, transmission, evolution, and pathogenesis. Future studies with additional medically relevant arboviruses should continue to probe this relationship.

## Supporting information

**S1 Fig. Removing the RNA extraction step increases the quantity of RNA detected from serum samples.** Serum samples from mice infected with Mayaro virus (MAYV) were used to assess the necessity for an RNA extraction step before one-step quantitative reverse-transcriptase PCR (qRT-PCR). qRT-PCR was performed using the NEB Luna Universal Probe kit with MAYV specific primers. For RNA extractions, the Zymo Viral DNA/RNA extraction kit was used. For testing without extraction, the samples were diluted 1:5 in PBS and then used directly for qRT-PCR. Viral RNA prepared from an infectious clone was serially diluted in ten-fold dilutions and used to generate a standard curve. The sample values were then fit to this standard curve and the volumes normalized to calculate the number of genome copies per mL of serum (genomes/mL). The whiskers represent the minimum and maximum points, the box limits represent the 25th and 75th percentile, and the horizontal line represents the median. Statistical comparison was performed using a two-tailed paired t-test.
(TIF)

**S2 Fig. Weights of mice following feeding on different diets.** Groups of C57BL/6N (A-B, 7–14 mice per group) or Balb/c (C, 5 mice per group) were fed a 45% fat (Obese), 5% protein (LP), or a control (Lean) diet for 8–10 weeks. Weights were measured weekly or bi-weekly following the initiation of feeding. Weights are plotted in grams. Statistical comparisons were

made using a two-way ANOVA with Dunnett's correction compared to the control group. The level of significance is represented as follows–ns (not significant) p>0.05, * P<0.05, **** p<0.0001. The **** represents the comparison between the lean and obese groups. The * represents the comparison between the lean and the low protein groups. The error bars represent standard error. Each graph represents data obtained from at least two independent experiments except for C, which was performed once. The results from one representative experiment are presented.
(TIF)

**S3 Fig. Weights of mice following infection with different alphaviruses.** Groups of C57BL/6N (A-B, 7–14 mice per group) or Balb/c (C, 5 mice per group) were fed a 45% fat (Obese), 5% protein (LP), or a control (Lean) diet for 8–10 weeks. Mice were then infected with Mayaro virus (MAYV, A), chikungunya virus (CHIKV, B), or Ross River virus (RRV, C). Weights were measured daily following infection. Weights are plotted in grams. The error bars represent standard error. Each graph represents data obtained from at least two independent experiments except for C, which was performed once. The results from one representative experiment are presented.
(TIF)

**S4 Fig. Histological lesions in mice with varying nutritional status.** Groups of C57BL/6N mice were fed either 45% fat (Obese), 5% protein (LP) or control (Lean) diet for 8–10 weeks. Mice were then infected with chikungunya virus (CHIKV). Nine days post-infection, the left hind footpad was collected and placed into formalin for fixation. The fixed tissues were then sectioned and stained with hematoxylin and eosin (H&E) and then scored in a blinded manner by two independent anatomic pathologists following the scoring guide presented by Hawman et al. 2013. Scale bar, 100 μM. Images are representative of 2 mice per group. The arrows represent areas of myositis.
(TIF)

**S5 Fig. Histological scores in mice with varying nutritional status.** Groups of C57BL/6N mice were fed either 45% fat (Obese), 5% protein (LP) or control (Lean) diet for 8–10 weeks. Mice were then infected with chikungunya virus (CHIKV). Nine days post-infection, the left hind footpad was collected and placed into formalin for fixation. The fixed tissues were then sectioned and stained with hematoxylin and eosin (H&E) and then scored in a blinded manner by two independent anatomic pathologists following the scoring guide presented by Hawman et al. 2013. The middle bar represents the median.
(TIF)

**S6 Fig. Organ titers of mice fed different diets.** Groups of C57BL/6N mice (9–10 mice per group) were fed either a 45% fat (Obese), 5% protein (LP), or control (Lean) diet for 8–10 weeks. Mice were then infected with Mayaro virus (MAYV) and 2-days post-infection the liver (A) and spleen (B) were collected. Viral titer was measured in the tissue homogenates by plaque assay. The whiskers represent the minimum and maximum points, the box limits represent the 25th and 75th percentile, and the horizontal line represents the median. Statistical analyses were performed by one-way ANOVA with Dunnett's correction compared to the control group. No statistically significant differences were found.
(TIF)

**S7 Fig. Sequencing coverage was even among the different groups tested.** Groups of C57BL/6N mice (7 mice per group) were fed either a 45% fat (Obese), 5% protein (LP), or control diet (Lean) for 8–10 weeks. Mice were then infected with Mayaro virus (MAYV) and blood was

collected two days later. RNA was isolated from the serum and next-generation sequencing libraries were prepared and then sequenced on an Illumina NextSeq 500. The FASTQ files were trimmed for adapter and low-quality sequences using BBDuk. The trimmed reads were then analyzed using the ViVAN pipeline with a coverage cutoff of >100x and a p-value filter of 0.05. From the output of ViVAN, we compared the sequencing coverage depth. The whiskers represent the minimum and maximum points, the box limits represent the 25th and 75th percentile, and the horizontal line represents the median. Statistical analyses were performed by one-way ANOVA with Dunnett's correction and compared to the control group.
(TIF)

**S8 Fig. Wild-type and marked reference virus fitness qRT-PCR is highly specific and sensitive.** 10-fold dilutions of RNA derived from either wild-type (WT) or the marked reference (REF) Mayaro virus (MAYV) infectious clone plasmids were made. RNA was then mixed at a 1:1 ratio between WT MAYV and REF MAYV to simulate a mixed infection. Probe-based qRT-PCR was then performed with either a probe specific for WT MAYV (A) or REF MAYV (B). Forty cycles of PCR were performed. The dotted line represents the limit of detection. Any point that did not produce a Ct value was given an arbitrary Ct value of 41.
(TIF)

**S9 Fig. Obese mice have reduced chronic RNA levels.** Groups of C57BL/6N mice (7 mice per group) were fed either a 45% fat (Obese), 5% protein (LP), or control (Lean) diet for 8–10 weeks. Mice were then infected with Mayaro virus (MAYV) and footpads were collected 65 days post-infection. Viral RNA was quantified using qRT-PCR. The whiskers represent the minimum and maximum points, the box limits represent the 25th and 75th percentile, and the horizontal line represents the median. Statistical analyses were performed by one-way ANOVA with Dunnett's correction compared to the control group. The level of significance is represented as follows—* $p < 0.05$.
(TIF)

**S1 Table. Composition of diets used in feeding experiments.**
(XLSX)

**S2 Table. Primers and probes used for real-time PCR.**
(XLSX)

## Acknowledgments

We are incredibly grateful to Yannis Michalakis and Renaud Vitalis for their stimulating discussions on genetic bottlenecks and thorough review of the manuscript. We thank Brandy Russell at the CDC for providing the virus strains used in these studies. We are grateful to Andrew Routh for his kind assistance with data analysis. We would also like to acknowledge Janet Webster for her thoughtful review of the manuscript. We much appreciate Yves Janin, Jean-Pierre Levraud, and Gabriella Passoni for their collaborations and enjoyable discussions. Finally, we are indebted to the animal care technicians and other technical staff at the Pasteur Institute, without whom this work would not have been possible.

## Author Contributions

**Conceptualization:** James Weger-Lucarelli, Marco Vignuzzi.

**Data curation:** James Weger-Lucarelli, Lucia Carrau, Laura I. Levi, Veronica Rezelj, Jérémy Boussier.

**Formal analysis:** James Weger-Lucarelli, Veronica Rezelj, Jérémy Boussier, Daniela Megrian, Tanya LeRoith, Marco Vignuzzi.

**Funding acquisition:** James Weger-Lucarelli, Marco Vignuzzi.

**Investigation:** James Weger-Lucarelli, Lucia Carrau, Laura I. Levi, Veronica Rezelj, Thomas Vallet, Hervé Blanc, Sheryl Coutermarsh-Ott, Tanya LeRoith.

**Methodology:** James Weger-Lucarelli, Laura I. Levi, Veronica Rezelj.

**Supervision:** Marco Vignuzzi.

**Writing – original draft:** James Weger-Lucarelli, Marco Vignuzzi.

**Writing – review & editing:** James Weger-Lucarelli, Lucia Carrau, Laura I. Levi, Veronica Rezelj, Marco Vignuzzi.

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
