## [Decision Letter · Decision Letter 0]

31 Jul 2019

Dear Dr. Weger-Lucarelli:

Thank you very much for submitting your manuscript "Host Nutritional Status Affects Alphavirus Virulence, Transmission, and Evolution" (PPATHOGENS-D-19-01146) for review by PLOS Pathogens. Your manuscript was fully evaluated at the editorial level and by independent peer reviewers. The reviewers appreciated the attention to an important topic but identified some aspects of the manuscript that should be improved.

We therefore ask you to modify the manuscript according to the review recommendations before we can consider your manuscript for acceptance. Your revisions should address the specific points made by each reviewer.

(1) A letter containing a detailed list of your responses to the review comments and a description of the changes you have made in the manuscript. Please note while forming your response, if your article is accepted, you may have the opportunity to make the peer review history publicly available. The record will include editor decision letters (with reviews) and your responses to reviewer comments. If eligible, we will contact you to opt in or out.

(2) Two versions of the manuscript: one with either highlights or tracked changes denoting where the text has been changed; the other a clean version (uploaded as the manuscript file).

We hope to receive your revised manuscript within 60 days or less. If you anticipate any delay in its return, we ask that you let us know the expected resubmission date by replying to this email.

[LINK]

Sincerely,

Richard J. Kuhn, PhD

Associate Editor

PLOS Pathogens

Mark Heise

Section Editor

PLOS Pathogens

Kasturi Haldar

Editor-in-Chief

PLOS Pathogens

orcid.org/0000-0001-5065-158X

Grant McFadden

Editor-in-Chief

PLOS Pathogens

orcid.org/0000-0002-2556-3526

Reviewer's Responses to Questions

**Part I - Summary**

Reviewer #1: This is an important piece of work that is aimed at understanding the effects of nutrition on alphavirus infections. The effects of host nutrition on virus infections in general is an understudied and under appreciated area. Therefore this paper has important implications for human health in an increasingly obese environment. However, there are some concerns with the paper that need to addressed.

Reviewer #2: In their manuscript, Weger-Lucarelli et. al. describe their study of nutritional influences on alphavirus infection mouse models. The authors find some interesting correlations between obesity/undernutrition and alphavirus infection. The paper is well written and clear and, overall, very good work. The topic is extremely novel and timely, especially with the global syndemic of malnutrition and endemic and emerging infectious diseases. However, some questions/issues exist and should be addressed in the context of these studies.

Reviewer #3: This study attempts to identify an important and previously unexplored link between host nutritional status and alphavirus infection. The study is straightforward and clearly outlined. The results are well presented. Statistics are acceptable. The discussion is well done with the exception of lines 400-409 which is essentially a repeat of the results. See below for specific comments:

**Part II – Major Issues: Key Experiments Required for Acceptance**

Reviewer #1: 1. Supplementary Table 1 was not included with the manuscript. This table described the nutrient content of the diets, so this is critical missing piece. I am assuming this was just an oversight by the authors

2. In the introduction, the authors state that they fed mice two high fat diets-45% and 61%, but the data presented only includes 45% high fat diets. Did they use the 61% diet? And if so, did the results mirror the 45% diet? This would be important information to know.

3. Authors should present actual weights (figures 1 and 2) of mice rather than just percent loss or percent gain of weight. It can be added as supplementary data, but it needs to be there. The readers have no idea if the starting weights were identical, and how much weight in grams were lost or gained. How many grams were the obese mice?

4. The authors mention diabetes in the discussion, but no metabolic markers were provided. What were the glucose levels of the mice? Were they, in fact, diabetic?

5. The authors measure leptin, but it’s unclear what the point is. It’s well known that leptin levels increase with adiposity. They did not see a leptin difference between low protein and control diet-what is the significance of this finding? Does it matter? This data could be removed, unless the authors provide more context. It seems to be just “thrown in” as a piece of data.

6. The legend for the stats on figure 1A is not clear. You have to read the text to understand the purpose of the one star that is shown on the data. The stats information should be clearly stated in the figure legend.

7. With the CHIKV challenge, why were weight actually increasing after infection for the non-obese groups?

8. For the pathology slides in Supp Fig 2, it would be helpful for the readers to use arrows to point out the inflammation, and to provide what histopathological score the photos represent.

9. The authors wanted to see if their results would be similar using a strain of mouse different from the B6 mice. This is a logical step, however, they also had to treat the mice with antibody to block type-I IFN-again, a major departure from the B6 mice. So this is not a good comparison between strains.

10. In the discussion section, they claim that the model of RRV led to 60% of the undernourished mice and 40% of the obese died from the infection, yet all the lean mice lived. However, in the results section, they state that 2 mice from the obese group and 3 mice from the LP group died-out of a total of 7 mice. This is not 60% mortality. And, they also state that the results were not statistically significant, so this does not rise to a level that needs to be discussed.

Reviewer #2: 1) Type of fat included in the diet can be very influential on the outcomes of experimental infection. How was this particular diet chosen as a proxy for biological relevance? Also, it may be good include a genetically obese model in the experimentation to show that obesity itself and not diet, can result in similar changes as observed in these studies. This would also be a very interesting question to help cover the influence of leptin on outcomes of these experiments as the authors have chosen to use this measure as a proxy of nutritional status.

2) Since adipose tissue is a highly vascularized organ, it is possible that the virus is being trapped in the extra adipose and that is why the authors do not observe differences (or even lowered) viremia? Was any attempt made to measure viral particles in other tissues (especially adipose) aside from the footpad and serum? Similar studies would be interesting for the persistence of viral RNA. Please consider doing this experimentation.

Reviewer #3: None

**Part III – Minor Issues: Editorial and Data Presentation Modifications**

Reviewer #1: Some paragraphs in the discussion need to be broken up for better readability.

Reviewer #2: 1) In the introduction, the authors refer to four different diets (Line 90), but only discuss three diets in the feeding methodology described on lines 140-141. Please consider revising for consistency. The remainder of the paper only refers to the 45% diet. If data is available for the 61% diet, please include in the paper.

2) Please give references in the methodology text for successful employment of these diets in previous studies.

3) Was any kind of clinical score (semi-quantitiative) taken during the mouse infections to monitor health of the animals? Hunching, coat quality, etc all could be monitored to increase knowledge of morbidity in these animals.

4) Lines 256: “weight gain” should just be “weight”?

5) Please show all data from C57 versus Balb/c experimentation, including leptin measurements. This data is especially important as these animals are generally less susceptible to dietary treatment that the more widely used C57Bl/6 mice.

6) The data from the serological viral titer determination and the luminescence studies seems to be contradictory. Was viremia/RNAemia measured at day 6 between obese and healthy models?

7) Is 2 days generally accepted as the period of time with the most changes in the viral quasispecies? This would be especially important to ask if a virus from a LP or obese mouse cause differential morbidity or mortality in a lean host aside from changes in fitness?

8) How do the genetic changes in the viruses correlate to pathogenicity of the viral species themselves aside from fitness?

9) While the LP diet is interesting (as it represents one of the main types of malnutrition, kwashiorkor), other types of malnutrition exist in areas endemic for these virsues including complete marasmus, caloric restriction, and micronutrient deficiencies. Can the authors further comment on how different types of undernutrition could further change the results of this study?

Reviewer #3: How much Protein is in the control diet? Currently, only obese and lean are described.

Figure 2: There is an interesting bi- or tri-phasic behavior in the foot pad swelling. Is there an explanation for this? It seems consistent in MAYV and CHIKV both. Also, line 255 says CHIKV and RRV, but only CHIKV is discussed. Was the data and discussion for RRV missed? It would be important to mention since E and F have a different mouse with RRV. Readers may be interested in the RRV C57bL/6 data for comparison.

Figure 4: B,C and D, labels difficult to see. What A represents is unclear. What ‘flux’ are we measuring?

PLOS authors have the option to publish the peer review history of their article (what does this mean?). If published, this will include your full peer review and any attached files.

Reviewer #1: No

Reviewer #2: No

Reviewer #3: No

---

## [Editor Report · Decision Letter 1]

17 Sep 2019

Dear Dr. Weger-Lucarelli,

We are pleased to inform that your manuscript, "Host Nutritional Status Affects Alphavirus Virulence, Transmission, and Evolution", has been editorially accepted for publication at PLOS Pathogens. 

Before your manuscript can be formally accepted and sent to production, you will need to complete our formatting changes, which you will receive by email within a week. Please note that your manuscript will not be scheduled for publication until you have made the required changes.

IMPORTANT NOTES

(1) Please note, once your paper is accepted, an uncorrected proof of your manuscript will be published online ahead of the final version, unless you’ve already opted out via the online submission form. If, for any reason, you do not want an earlier version of your manuscript published online or are unsure if you have already indicated as such, please let the journal staff know immediately at plospathogens@plos.org.

(2) Copyediting and Proofreading: The corresponding author will receive a typeset proof for review, to ensure errors have not been introduced during production. Please review the PDF proof of your manuscript carefully, as this is the last chance to correct any errors. Please note that major changes, or those which affect the scientific understanding of the work, will likely cause delays to the publication date of your manuscript. 

(3) Appropriate Figure Files: Please remove all name and figure # text from your figure files. Please also take this time to check that your figures are of high resolution, which will improve the readbility of your figures and help expedite your manuscript's publication. Please note that figures must have been originally created at 300dpi or higher. Do not manually increase the resolution of your files. For instructions on how to properly obtain high quality images, please review our Figure Guidelines, with examples at: http://journals.plos.org/plospathogens/s/figures.

(4) Striking Image: Please upload a striking still image to accompany your article if one is available (you can include a new image or an existing one from within your manuscript). Should your paper be accepted, this image will be considered for our monthly issue image and may also appear on our website to feature your article. Please upload this as a separate file, selecting "striking image" as the file type upon upload. Please also include a separate "Other" file with a caption, including credits and any potential copyright information. Please do not include the caption in the main article file. If your image is from someone other than yourself, please ensure that the artist has read and agreed to the terms and conditions of the Creative Commons Attribution License at http://journals.plos.org/plospathogens/s/content-license. Please note that PLOS cannot publish copyrighted images.

(5) Press Release or Related Media: If your institution or institutions have a press office, please notify them about your upcoming paper at this point, to enable them to help maximize its impact. If they will be preparing press materials for this manuscript, please inform our press team in advance at plospathogens@plos.org as soon as possible. We ask that you contact us within one week to plan ahead of our fast Production schedule. If you need to know your paper's publication date for related media purposes, you must coordinate with our press team, and your manuscript will remain under a strict press embargo until the publication date and time. This means an early version of your manuscript will not be published ahead of your final version. 

(6)  PLOS requires an ORCID iD for all corresponding authors on papers submitted after December 6th, 2016. Please ensure that you have an ORCID iD and that it is validated in Editorial Manager.  To do this, go to ‘Update my Information’ (in the upper left-hand corner of the main menu), and click on the Fetch/Validate link next to the ORCID field.  This will take you to the ORCID site and allow you to create a new iD or authenticate a pre-existing iD in Editorial Manager

(7) Update your Profile Information: Now that your manuscript has been provisionally accepted, please log into Editorial Manager and update your profile, if needed. Go to https://www.editorialmanager.com/ppathogens, log in, and click on the "Update My Information" link at the top of the page. Please update your user information to ensure an efficient production and billing process. 

(8) LaTeX users only: Our staff will ask you to upload a TEX file in addition to the PDF before the paper can be sent to typesetting, so please carefully review our Latex Guidelines http://journals.plos.org/plospathogens/s/latex in the meantime.

(9) If you have associated protocols in protocols.io, please ensure that you make them public before publication to guarantee immediate access to the methodological details.

Best regards,

Richard J. Kuhn, PhD

Associate Editor

PLOS Pathogens

Mark Heise

Section Editor

PLOS Pathogens

Kasturi Haldar

Editor-in-Chief

PLOS Pathogens

orcid.org/0000-0001-5065-158X

Grant McFadden

Editor-in-Chief

PLOS Pathogens

orcid.org/0000-0002-2556-3526

The authors have responded to each of the three reviewers' questions and concerns. Significant improvements and clarification were made as a result. The response was appropriate and the manuscript has been improved as a consequence and is now acceptable for publication.
---

## [Editor Report · Acceptance letter]

2 Oct 2019

Dear Dr. Weger-Lucarelli,

We are delighted to inform you that your manuscript, "Host Nutritional Status Affects Alphavirus Virulence, Transmission, and Evolution," has been formally accepted for publication in PLOS Pathogens.

Best regards,

Kasturi Haldar

Editor-in-Chief

PLOS Pathogens

orcid.org/0000-0001-5065-158X

Grant McFadden

Editor-in-Chief

PLOS Pathogens

orcid.org/0000-0002-2556-3526